# Toward Trustworthy Neural Program Synthesis

## Abstract

We develop an approach to estimate the probability that a program sampled from a large language model is correct. Given a natural language description of a programming problem, our method samples both candidate programs as well as candidate predicates specifying how the program should behave. This allows learning a model that forms a well-calibrated probabilistic prediction of program correctness. Our system also infers the which predicates are useful to explain the behavior of the generated code, and humans preferred these in a human study over raw language model outputs. Our method is simple, easy to implement, and maintains state of the art generation accuracy results.

## 1 Introduction

Many AI systems attempt to resolve bugs and other software engineering problems by automatically generating patches purportedly solving GitHub issues **devin openhands swebench and a few other agents** These LLM systems are good, but not perfect. These neural systems are good, but not perfect. Suppose 75% of the time, such systems propose a correct fix to the GitHub issue. The other 25% of the time, they produce plausible looking code containing subtle bugs. Would you use this system?

Many engineers would be reluctant to use such a system, because it fails to build trust with the user. When it fails, it cannot detect its own failure. When it succeeds, it doesn't present a human-understandable explanation of why its program behaves as intended. In this paper we seek steps towards rectifying this lack of trust by building natural-language conditioned program synthesizers that are more trustworthy in several complimentary ways:

- **Calibration**: We want systems that, when they cannot solve a programming problem, simply return no answer, rather than return a (possibly subtly) incorrect program. We conjecture that it is better to fall back on the human programmer, rather than risk introducing bugs. Contrast with natural language translation: Unlike natural language, programs are brittle, and more time-consuming to look into and understand. And debugging bad code, unlike proofreading language, can be harder than just writing it yourself. We do this by having a classifier predict whether the program is correct, and making the classifier well-calibrated (Platt et al., 1999; Kuhn et al., 2023).

- **Explainability**: To help humans understand the output of a neural network that is writing code, we want a system that can explain its outputs by generating informative and interpretable checks on program behavior. We propose a characterization of what makes an explanation of program behavior 'good', and validate in a human user study that this characterization produces better explanations than a raw large language model by itself.

- **Accuracy**: Ideally, trustworthy systems should be more accurate, solving more programming problems. This goal might seem to be in tension with the previous two. Surprisingly, we find our methods also boost overall accuracy on natural language to code generation problems.

Our high-level approach has a neural network propose candidate program solutions and independently propose predicates that correct solutions should satisfy, such as Input/Output tests, known as **specifications** ('specs', Fig. 1). Although specs can refer to a broad range of ways to specify the behavior of programs, here we only consider two kinds: (1) input-output test cases, and (2) parameterized logical relation tests which execute the program and check some relation between input and output. In general, a spec can be any mechanically checkable property. We check the programs

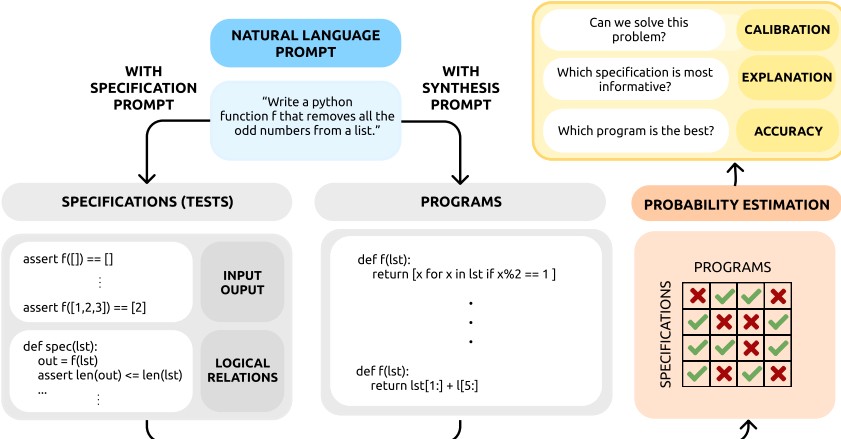

Figure 1: Our `speculyzer` system inputs a natural language description of a programming problem. It uses large language models to independently sample candidate programs, and candidate specifications of what the program should do. Because natural language is informal, we cannot mechanically check programs against it, but logical relations and input-outputs can be mechanically checked against. The result of this validation step is fed to a learned model which predicts whether the problem can be solved; if so, which program is correct; and which specs best explain the behavior of the program, am would be useful for judging whether that program is correct or incorrect.

against the specs, and learn to use this checking to predict if the system knows how to solve the problem at all, and if so, which program(s) are probably the right solution. Intuitively, we ask the language model to 'check its work' by generating specs. We call our approach `speculyzer`, short for 'Specification Synthesizer', because in addition to synthesizing programs, it synthesizes specs.

Our work makes the following contributions:

1. **Calibration** A method to give a well-calibrated probabilistic estimate of whether a program is correct which enables analysis of metrics connected to trust and safety

2. **Explainability** A method for identifying the specifications most likely be useful to humans as an explanation of program behavior, and a validation of that approach in a human study

3. **Accuracy** Demonstration that the above contributions do not impair the overall accuracy of programs synthesizers, and can sometimes let them solve more problems overall

## 2 RELATED WORK

**Program synthesis.** Automatically constructing software has been a longstanding goal of computer science (Manna & Waldinger, 1979; Gulwani et al., 2017). Classic program synthesizers input a formal specification, and then either search or logically derive a program guaranteed to satisfy that formal specification (Alur et al., 2013).

**Large language models for source code.** Our work uses large language models for source code (Chen et al., 2021; Austin et al., 2021). These neural networks generate source code conditioned or 'prompted' by a mix of code and natural language (the natural language is usually represented as code comments). Such models are typically implemented as large transformers (Vaswani et al., 2017; Brown et al., 2020).

Following the introduction of large transformer-based language models for source code, there has been work on how to boost the accuracy of those models. Here, accuracy means the probability of sampling a correct program conditioned on a natural-language prompt. Accuracy is often measured by functional correctness with the *pass@k* metric, which considers drawing $k$ IID samples from the language model and testing if any samples pass a set of holdout test cases. Toward boosting *pass@k*, researchers have considered *clustering* sampled programs according to the outputs they produce on test inputs (Shi et al., 2022; Li et al., 2022). For example, AlphaCode prioritizes large 'clusters' of

samples with the exact same input-output behavior (Li et al., 2022), effectively reranking the samples from the language model according to how likely they are to solve the task. Another strategy is to train a second neural network to predict program correctness (Inala et al., 2022).

The closest work to ours is CodeT (Chen et al., 2023a), which also generates programs as well as input-output test cases, with the goal of boosting *pass@k*. The difference between our systems is that we designed `speculyzer` to build trust in a variety of ways by synthesizing specifications, centered around first forming well-calibrated probability estimates, only boosting *pass@k* as a side effect. We also incorporate input-output test cases as a special case of specs in general.

Engineering safe, trustworthy language models has received considerable attention by the AI safety (Thoppilan et al., 2022) and AI alignment communities (Kadavath et al., 2022). These works find that one can train classifiers which predict the truthfulness or safety of language outputs by inspecting the hidden activations of the model or even by simply 'asking' the model if its output is correct or safe. We see this family of efforts as complementary: For programs, it is possible to formally specify correctness properties, which is not generally true in NLP, so we focus on formal properties (specifications) here. Nonetheless, one can train statistical predictors of program correctness (Inala et al., 2022). Broadly however, we think that program synthesis offers unique opportunities for building trust through symbolic methods. Although statistically reranking language model outputs via a second neural network improves raw performance, we believe it is a suboptimal trust-builder: an inscrutable neural network cannot guarantee the correctness of another inscrutable network. Here we advocate that properties which are symbolically checkable and human-comprehensible should play a role, and examine certain specifications as basic examples of such properties.

## 3 METHODS

Given a natural-language prompt describing a programming problem, our goal is to produce a well-calibrated estimate probability of each program sample from language model being correct. Our approach independently samples a set of candidate programs $\mathcal{P}$ and a set of candidate specs $\mathcal{S}$. Specs can be either input-output testcases, or logical relations (Fig. 1). We write $\mathcal{T}$ for the set of test cases and $\mathcal{R}$ for the set of logical relations, so $\mathcal{S} = \mathcal{T} \cup \mathcal{R}$. Each program $p \in \mathcal{P}$ is checked against each spec $s \in \mathcal{S}$, and basic statistics of program-spec agreement are computed. These statistics are aggregated by a learned model into an estimated probability that the program is correct. Programs whose probability falls below a threshold are discarded. Any remaining programs are sorted by probability and returned to the user as possible solutions, together with certain specs they pass. Returning specifications allows the user to check that the code has the intended behavior. This architecture lets the system learn how to predict when it cannot solve a problem, and also learn to rank candidate solutions and their corresponding specs.

### 3.1 SAMPLING PROGRAMS AND SPECS

Given a string `prompt` describing a programming problem, we sample $n = 100$ candidate programs (the set $\mathcal{P}$) and candidate specs (the set $\mathcal{S}$). Both sets are sampled using a pretrained language model, which can probabilistically generate program source code conditioned on a prompt. We write $P^{\text{LM}}(\cdot|\texttt{prompt})$ for the conditional distribution over programs, given `prompt`. If a program $p \in \mathcal{P}$, then $p \sim P^{\text{LM}}(\cdot|\texttt{prompt})$. To sample specs, we deterministically transform the prompt as in Fig. 2 and Appendix, then draw iid samples from the language model to construct relations $\mathcal{R}$ and input-output test cases $\mathcal{T}$.

One motivation for generating logical relations is that large language models are famously bad at predicting the output of a novel program Nye et al. (2021). However, given a generic input, they can produce a variety of reasonable constraints on that the output should obey, if they are prompted in a program-of-thought style Chen et al. (2022); Gao et al. (2023). Logical relations also resemble unit test harnesses, like those used in property based testing, Fink & Bishop (1997) which are likely part of the model's training data. We therefore suspected that although input-outputs are the superior form of test for typical programming tasks, logical relations could serve the long tail of novel tasks that the LLM cannot reliably predict outputs for.

Figure 2: Our systems use three types of prompts to generate programs, input-output tests, and logical relations. Prompts in gray and completion in bold.

## 3.2 CHECKING SPECS AGAINST PROGRAMS

To judge the correctness of a program $p$, we would like to know whether each specification $s \in \mathcal{S}$ is actually true of $p$, notated $p \models s$. If the specification $s$ is an input-output test, this checking is trivial: the program $p$ can be run on the input and checked to see if it yields the desired output. But if $s$ is a logical relation, checking if $p \models s$ over the entire input space becomes undecidable. Thus we require an approximate validation approach for logical relations. As a heuristic approximation we ask the language model to generate a few candidate inputs on which to run the relational test, effectively using the language model as a fuzzer. This causes us to overapproximate the set of relations a program satisfies. Notionally we write $p \vdash_T s$ to mean that spec $s$ is inferred to be true of program $p$ according to testing methodology $T$. If $T = IO$, then $p \models s$ iff $p \vdash_{IO} s$. If $s$ is a relation, we instead have $p \models s$ implies $p \vdash_{Rel} s$. To avoid confusion, we always show the concrete inputs on which a logical relation was tested.

## 3.3 SCORING AND ANALYZING TEST COVERAGE

Given programs $\mathcal{P}$ and specs $\mathcal{S}$, we produce an estimated probability for each $p \in \mathcal{P}$ that $p$ is correct. Assuming, on average, specs correctly formalize the informal natural-language intention, satisfying more specs should increase our confidence in a program. Additionally, if many sampled programs exhibit identical behavior on the specs, then we should increase our confidence in those programs, because this indicates high marginal probability of that behavior under $P^{\text{LM}}(\cdot|\texttt{prompt})$. This 'clustering' of candidate solutions according to their execution behavior, and prioritizing large clusters, has been successfully used by (Li et al., 2022; Lewkowycz et al., 2022; Shi et al., 2022). It is also related to *observational equivalence* from classic program synthesis (Udupa et al., 2013), which treats programs as identical if they have the same outputs on tests.

Therefore, we estimate the probability of correctness for each program $p \in \mathcal{P}$ using a logistic regressor over features $\phi(p, \mathcal{P}, \mathcal{S})$ and learned parameters $\theta$. The features include **i/o pass rate** (input-output specs passed), **relation pass rate** (logical-relation specs passed, under fuzzing), **cluster size** (# of other programs satisfying the same specs), the ordinal rank of the preceding features and the normalized log probability of the sampled programs:

$$\text{score}_\theta(p|\mathcal{P}, \mathcal{S}) = \text{Sigmoid}\left(\theta \cdot \phi(p, \mathcal{P}, \mathcal{S}) + \theta_0\right) \qquad (1)$$

where the components of $\phi(p, \mathcal{P}, \mathcal{S})$ include the following and their ordinal ranks:

$$\text{IOPass}(p, \mathcal{P}, \mathcal{T} \cup \mathcal{R}) = \frac{1}{|\mathcal{T}|} \sum_{s \in \mathcal{T}} \mathbb{1}\left[p \vdash_{IO} s\right] \qquad \text{RelPass}(p, \mathcal{P}, \mathcal{T} \cup \mathcal{R}) = \frac{1}{|\mathcal{R}|} \sum_{s \in \mathcal{R}} \mathbb{1}\left[p \vdash_{Rel} s\right]$$

$$\text{ClusterSize}(T, p, \mathcal{P}, \mathcal{S}) = \sum_{p' \in \mathcal{P}} \prod_{s \in \mathcal{S}} \mathbb{1}\left[(p \vdash_T s) = (p' \vdash_T s)\right] \qquad \text{where } T \in \{IO, Rel\}$$

$$\text{NormLogProb}(p) = \frac{1}{\text{len}(p)} \log P^{\text{LM}}(p|\texttt{prompt})$$

We fit $\theta$ via maximum likelihood on a corpus $\mathcal{D}$ containing triples $\langle \mathcal{P}, \mathcal{S}, \mathcal{G} \rangle$ of programs $\mathcal{P}$ and specifications $\mathcal{S}$, both sampled from the same problem, and ground-truth testcases $\mathcal{G}$, which serve as a proxy for program correctness. The ground truth test cases $\mathcal{G}$ are assumed to be unavailable at test time, because our goal is synthesis from informal specifications like natural language. We use gradient ascent to maximize the log likelihood $\mathcal{L}$.

$$\mathcal{L} = \sum_{\langle \mathcal{P}, \mathcal{S}, \mathcal{G} \rangle \in \mathcal{D}, p \in \mathcal{P}} \left( \mathbb{1}\left[p \vdash_{IO} \mathcal{G}\right] \log \text{score}_\theta(p|\mathcal{P}, \mathcal{S}) + \mathbb{1}\left[p \nvdash_{IO} \mathcal{G}\right] \log \left(1 - \text{score}_\theta(p|\mathcal{P}, \mathcal{S})\right) \right)$$

### 3.4 TEST TIME METRICS

**Precision-Recall.** We seek high *precision* without sacrificing *recall*. High precision means that when the system suggests a program, it is probably correct. High recall means a correct program achieves the top rank: In other words, the system can solve a lot of programming problems, though it might make more mistakes in the process. The tradeoff between precision and recall can be tuned by a thresholding parameter, $\tau$. A candidate program is discarded if its score falls below the threshold $\tau$. If all programs are discarded, the system declines to provide an output for the programming problem, and otherwise the system outputs a ranked list of programs sorted by score. We define Precision and Recall as follows, which respectively measure (1) if a correct program is top-ranked whenever any program scores above $\tau$ and (2) how often a correct program scoring above $\tau$ is the top ranked.

$$\text{Precision@}k = \frac{\text{TruePositives@}k}{\text{PredictedPositives}} \qquad \text{Recall@}k = \frac{\text{TruePositives@}k}{\text{ActualPositives}}$$

$$\text{TruePositives@}k = \sum_{\langle \mathcal{P}, \mathcal{S}, \mathcal{G} \rangle \in \mathcal{D}} \mathbb{1}\left[\begin{array}{l} \exists p \in \mathcal{P} : p \vdash_{IO} \mathcal{G} \wedge \\ \tau \leq \text{score}_\theta(p|\mathcal{P}, \mathcal{S}) \wedge \\ p \in \text{top-k}_{p' \in \mathcal{P}} \text{score}(p'|\mathcal{P}, \mathcal{S}) \end{array}\right]$$

$$\text{PredictedPositives} = \sum_{\langle \mathcal{P}, \mathcal{S}, \mathcal{G} \rangle \in \mathcal{D}} \mathbb{1}\left[\exists p \in \mathcal{P} : \tau \leq \text{score}_\theta(p|\mathcal{P}, \mathcal{S})\right]$$

$$\text{ActualPositives} = \sum_{\langle \mathcal{P}, \mathcal{S}, \mathcal{G} \rangle \in \mathcal{D}} \mathbb{1}\left[\exists p \in \mathcal{P} : p \vdash_{IO} \mathcal{G}\right]$$

We sweep possible values for $\tau$ to compute a precision-recall curve. Generically, there is no 'true' best trade-off between these desiderata.

**Pass rate.** The *pass@k* metric (Austin et al., 2021; Chen et al., 2021) measures the probability of $k$ samples from $P^{\text{LM}}(\cdot|\texttt{prompt})$ passing the ground-truth test cases, $\mathcal{G}$:

$$pass@k = \mathbb{E}_{p_1 \dots p_k \sim P^{\text{LM}}(\cdot|\texttt{prompt})} \mathbb{1}\left[\exists p_i : p_i \vdash_{IO} \mathcal{G}\right]$$

Note that *pass@k* is proportional to ActualPositives: The (fraction of) problems where there is at least one correct answer in the sampled programs.

It is also conventional to combine *pass@k* with a scoring function that reranks the sampled programs. This generalizes *pass@k* to *pass@k,n*, which measures the probability that, after generating $n$ candidate programs, a correct one is in the top-$k$ under our scoring function:

$$pass@k,n = \mathbb{E}_{\langle \mathcal{P}, \mathcal{T}, \mathcal{G} \rangle \sim \mathcal{D}} \mathbb{1}\left[\begin{array}{l} \exists p \in \text{top-k}_{p' \in \mathcal{P}} \text{score}_\theta(p'|\mathcal{P}, \mathcal{T}) \\ \text{where } p \models \mathcal{G} \end{array}\right]$$

## 4 RESULTS

We study our approach on two popular datasets while using Codex models (Chen et al., 2021)[1] of different sizes, seeking to answer the following research questions:

- Is the classifier well-calibrated? If so, how trustworthy can we make the system (precision), and how much does that require sacrificing coverage (recall)?
- How can we use the synthesized specifications to act as human-interpretable explanations of the behavior of the programs constructed by the language model?
- How does our learned reranking impact raw rate of success (*pass@k,n*)?

We evaluate on programming problems from the Mostly Basic Python Problems (MBPP: Austin et al. (2021), sanitized version) and HumanEval datasets (Chen et al., 2021). Each of these datasets contains natural language descriptions of programming problems, and holdout tests to judge program correctness. An important difference between them is that HumanEval sometimes includes example input-outputs as part of the natural language description, while MBPP does not. Having I/O examples in the problem description makes spec generation easier: some specs are given for free. On the other hand, humans sometimes spontaneously mix natural language and examples (Acquaviva et al., 2021). Therefore, using both MBPP and HumanEval gives a more robust evaluation, but we note this qualitative difference between them. We gives further experimental setup details, such as hyperparameters and example prompts in the Appendix.

### 4.1 CALIBRATION, PRECISION, AND RECALL

Trustworthy systems should avoid predicting any programs at all when they cannot solve a problem. Doing so increases precision, the fraction of outputs which are correct, but at the expense of recall, the fraction of solvable problems where we output a correct solution. Fig. 3 illustrates how one can adjust this trade-off. For example, we can achieve 100% precision on HumanEval (*zero* error rate), in exchange for dropping our recall from 93.4% to 44.4%. Note this zero error rate does not come from our learned score function memorizing the data: we use cross validation to test each program using weights trained on other folds. Less extreme tradeoffs are possible, such as 90% precision in exchange for 90.7% recall.

Striking favorable points on these tradeoffs is, in theory, a result of having a *well-calibrated model*: whenever our probabilistic scoring function assigns probability $x\%$ to a program being correct, then approximately $x\%$ of the time, that program should actually be correct. We confirm this calibration property in Fig. 4, also finding that raw log likelihoods from the language model are substantially less well-calibrated. Calibration allows tuning the threshold $\tau$ to achieve a desired precision, because the free parameter $\tau$ acts as a threshold on the probability of correctness needed to output a program.

Fig. 3 also shows that our full method typically performs best on precision/recall statistics in the most realistic setting, namely using the largest 'Davinci' model. Nonetheless, using just input-output specs, or just logical relations, can be effective on their own, as both outperform the random baseline which assigns a uniform probability to all programs sampled from the language model.

### 4.1.1 FURTHER APPLICATION: MIXING LANGUAGE MODELS OF DIFFERENT SIZES

Large language models are expensive and energy-intensive, but often come in smaller, cheaper sizes. In theory, predicting whether a language model can solve a problem should allow us to efficiently multiplex between a small cheap model and a large powerful model by only delegating to the large model those problems which we predict cannot be solved by the smaller model. We use our learned classifier to perform exactly this multiplexing, with Cushman-size Codex as the cheap model and Davinci-size as the powerful model as illustrate in Fig. 5

Using our probabilistic model to switch between small and large LLMs, we can approximately halve the number of queries to the large model while maintaining a similar number of solved programming

---

[1]These experiments were conducted during a period when Codex models were widely used in the literature, allowing proper comparison with concurrent work like CodeT. Additionally, using older models helps avoid benchmark contamination in HumanEval and MBPP.

| | HumanEval | | | MBPP | | |
|---|---|---|---|---|---|---|
| | AUC | max F1 | R@ P=.9 | AUC | max F1 | R@ P=.9 |
| Davinci (Largest Codex model) | | | | | | |
| full | **0.91** | **0.91** | **0.91** | **0.74** | **0.80** | **0.43** |
| IO only | 0.86 | 0.88 | 0.83 | 0.69 | 0.75 | 0.41 |
| rels only | 0.66 | 0.73 | 0.27 | 0.69 | 0.77 | 0 |
| random | 0.15 | 0.39 | 0 | 0.23 | 0.48 | 0 |
| Cushman (Smaller Codex model) | | | | | | |
| full | **0.67** | **0.72** | 0.28 | **0.57** | **0.65** | 0.18 |
| IO only | 0.66 | 0.71 | **0.33** | 0.55 | 0.63 | **0.25** |
| rels only | 0.34 | 0.44 | 0.14 | 0.51 | 0.61 | 0.14 |
| random | 0.08 | 0.28 | 0 | 0.16 | 0.40 | 0 |

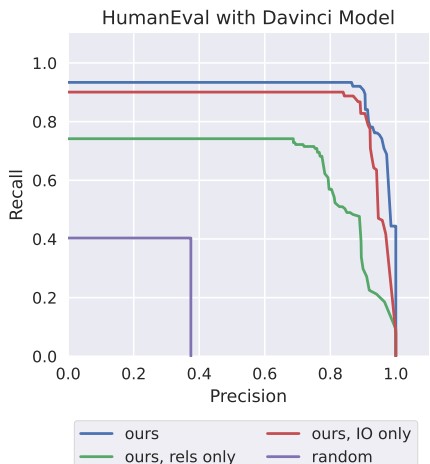

Figure 3: Left: Statistics of these curves, measuring Area Under Curve (AUC), max F1 (harmonic mean of precision and recall), recall in the high-trust regime: R@P=.9 is recall when precision=90%. The right figure gives further results.

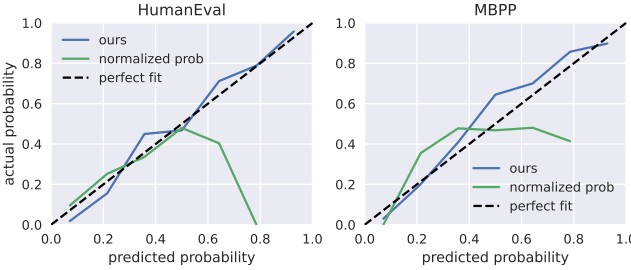

Figure 4: Comparing the calibration of our probabilistic scoring function against raw Codex(Davinci) log likelihood. Normalized probability, $Pr[\texttt{tokens}|\texttt{prompt}]^{\frac{1}{\texttt{tokens\_length}}}$, is the common metric used for scoring LLM samples (Lin et al., 2022; Si et al., 2022).

problems (Fig. 5). Contemporaneous work Chen et al. (2023b) has also considered similar cascading of language models, but required bespoke algorithms to learn the multiplexing policy. Here we show that a decent mixing of language model sizes can be accomplished 'for free' when we train a well-calibrated classifier to predict whether a program is correct.

## 4.2 SPECS AS EXPLANATIONS FOR PROGRAM BEHAVIOR

No natural language program synthesizer will always produce correct programs: Therefore, the system needs to explain what a synthesized program $p$ computes, so that the user can confidently accept or discard it. `speculyzer` does this by outputting a specification that is true about $p$, while being maximally informative as to $p$'s behavior. Below we formalize what it means to be 'maximally informative', and describe a study with human participants that shows that they prefer the explanations generated by our approach, compared to the raw output of the LLM.

Whenever `speculyzer` ranks $p^* \in \mathcal{P}$ as the best solution to a problem, it selects a spec $s^* \in \mathcal{S}$ to communicate the behavior of $p^*$. The spec $s^*$ must be a true fact about $p^*$, so $p^* \vdash s^*$, but should also constrain the behavior of $p^*$. For example, the specification $\forall x : p^*(x) = p^*(x)$ is vacuously true for any $p^*$, and so makes a poor explanation, despite holding for the ground-truth program. As an example of a good explanation, if the program $p^*$ is incorrect, then the spec $s^*$ *should also be incorrect*, in the sense that it is not true of the ground-truth program, despite holding for $p^*$.

We formalize this as a rational-communication model of program synthesis (Pu et al., 2020), which means first defining a joint probability distribution over programs and specifications: $\mathbb{P}[p, s] \propto \mathbb{1}[p \vdash s]\,\mathbb{1}[p \in \mathcal{P}]\,\mathbb{1}[s \in \mathcal{S}]$. Then, we score each specification $s$ by the conditional probability of

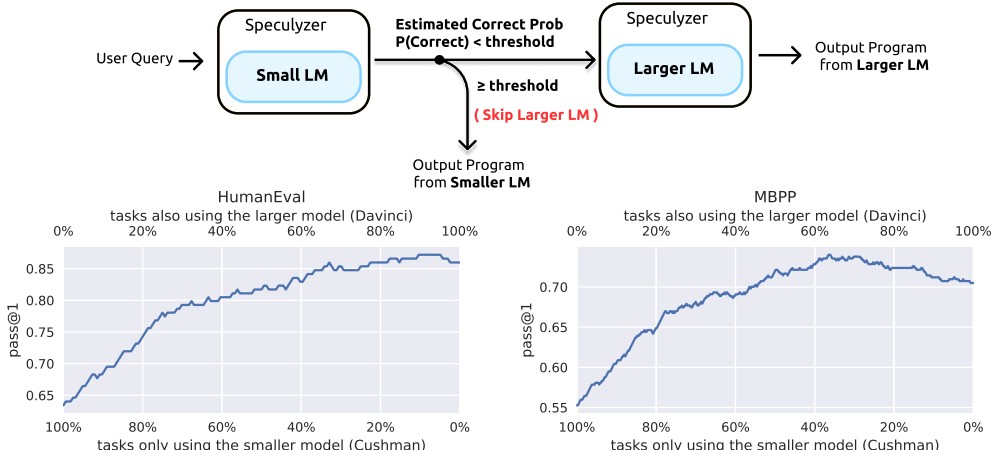

Figure 5: Mixing small cheap models and large powerful models by delegating to the large model only those problems that our approach thinks that the small model cannot solve. There are some MBPP problems where small model beats the big one. Our method multiplexes correctly for those.

$p^*$ given $s$, i.e. $\mathbb{P}[p^*|s]$. Applying Bayes' Rule and simplifying, we find that this is equivalent to ranking specs by how few other programs satisfy them, i.e. their selectivity:

$$s^* = \arg\max_{s \in \mathcal{S}} \mathbb{P}[p^*|s] = \arg\min_{\substack{s \in \mathcal{S} \\ p^* \vdash s}} \sum_{p \in \mathcal{P}} \mathbb{1}[p \vdash s]$$

To confirm that this is an effective method of scoring specifications, we recruited 22 human programmers, and ran an IRB-approved study where we showed them 8 programs synthesized by our system, together with the top-ranked spec, bottom ranked spec, and a random spec (Fig. 6). Every spec was a predicate that the program actually satisfied, and was the output of code-davinci-002. Study participants rated each of the specifications on a 1-7 Likert scale, and were instructed to score how helpful the specification is in communicating whether the program is correct or incorrect. As shown in Fig. 7, participants preferred the top-ranking specification over raw specs sampled from the LLM ($p < .05$, using a two-tailed $t$-test), and also dispreferred bottom-ranking specifications over the raw LLM output ($p < .05$, using the same statistical test). However, the overall effect size was modest (Cohen's $d = 0.14$), suggesting improving either the spec ranking heuristic or the underlying specs themselves (e.g., by fine-tuning the LLM) could be important in future work. These results, however, clearly establish that the above probabilistic model is better than the LLM on its own for generating interpretable explanations about program behavior that allow humans to decide whether a program is correct or not.

### 4.3 SPECULYZER FOR RANKING PROGRAMS

So far we have shown how our system forms well-calibrated probability estimates of program correctness, which allows predicting whether a problem can be solved or not, in addition to inferring what the best program might be. This is a harder, more general problem than reranking sampled programs to improve the accuracy of program synthesis. How well then do our probability estimates serve as a ranking function, and does solving this harder problem sacrifice performance on the easier challenge of reranking sampled programs? To answer this question, we measure *pass@1,100* using our scoring function to rank sampled programs. This means we sample 100 candidate programs, use our classifier to assign them probabilities, and then return the 1 program with the highest probability. We assess our system using repeated 5-fold cross validation and present the results in Figure 8. We can contrast with raw draws from the LLM (random), the state of the art approach to ranking samples from language models over code (CodeT), and a hypothetical Oracle that always chooses a correct program, if it exists.

Overall, `speculyzer` achieves 85.7% *pass@1* on HumanEval and 70.5% *pass@1* on MBPP. These are better than comparable published numbers for reranking methods (Chen et al., 2023a;

```
PROGRAM

def derivative(xs: list):
    """ xs represent coefficients of a polynomial.
    xs[0] + xs[1] * x + xs[2] * x^2 + ....
     Return derivative of this polynomial in the same form.
    >>> derivative([3, 1, 2, 4, 5])
    [1, 4, 12, 20]
    >>> derivative([1, 2, 3])
    [2, 6]
    """
    return [x * i for i, x in enumerate(xs) if i != 0]
```

```
TOP LOGICAL RELATION

def test_derivative(xs: list):
    """ Given an input `xs`, test whether the function `derivative`
is implemented correctly.
    """
    ys = derivative(xs)
    assert len(ys) == len(xs) - 1
    for i in range(len(ys)):
        assert ys[i] == xs[i+1] * (i + 1)

# run `test_derivative` on a new testcase
test_derivative([3, 1, 2, 4, 5])
```

```
RANDOM LOGICAL RELATION

def test_derivative(xs):
    """ Test function derivative().
    """
    # TODO
    pass

# run `test_derivative` on a new testcase
test_derivative([2, 3, 4, 10, -12])
```

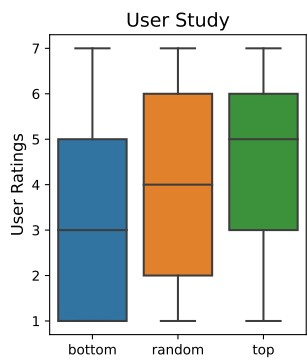

Figure 7: User ratings (1-7 Likert scale) for program specifications: top-ranked, random-ranked, and bottom-ranked. Participants prefer top ranked specs over random ones, and random ones over bottom ranked specs.

Figure 6: Program and specifications generated by language model and then reranked by our scoring function.

Inala et al., 2022; Zhang et al., 2022; Ni et al., 2023). This shows that we can achieve calibration via accurate probability estimates without sacrificing raw accuracy, and can even improve the state-of-the-art CodeT reranking method.

| Method | Random | Oracle | Ours | CodeT* |
|---|---|---|---|---|
| | | HumanEval | | |
| cushman | 25.0 | 81.1 | **63.3** ±0.59 | 58.6 |
| davinci | 37.5 | 92.1 | **85.7** ±0.73 | 74.8 |
| | | MBPP | | |
| cushman | 35.6 | 78.9 | **55.6** ±0.35 | 55.4 |
| davinci | 44.6 | 84.8 | **70.5** ±0.24 | 67.7 |

Figure 8: Pass@1,100: Calibration does not come at the expense of accuracy. CodeT results from Chen et al. (2023a)

## 5 CONTRIBUTIONS AND OUTLOOK

We have contributed a program synthesizer that learns to predict when it cannot solve a problem and selects specs that communicate what each program does. We intend for these to increase the trust and safety of neural program synthesis and to serve as a modest step toward program synthesizers that could better collaborate with software engineers. These advances do not come at the expense of raw accuracy, and also facilitate mixing neural networks of different sizes.

Many directions remain open. The idea of formal specifications as a liaison between programs and informal natural language opens up the possibility of using richer kinds of specs and verifiers, tapping years of effort from the programming languages community (D'silva et al., 2008; Baldoni et al., 2018), at least if we can interface such formalisms with large language models. Replacing program execution with sophisticated verification would also mitigate the aforementioned security concerns. Another direction is integration with advanced HCI for program synthesis, such as Peleg et al. (2020), which develops powerful human interaction paradigms for program synthesis.

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

## A  EXPERIMENTAL SETUP

**Sampling from language models.** We used Codex models to draw samples using a max-token size of 300 for our generation of programs and IO tests for both HumanEval and MBPP. For logical relation tests, we use 768 and 512 for the Davinci and Cushman models respectively. We used `"\ndef"`, `"\n#"`, `"\nclass"`, `"\nif"`, and `"\nprint"` as stop tokens for our generation of programs and input-output test cases, and we used `"\n# Problem {number}"` as the stop token for our generation of the logical relations test cases where `number` is set according to the number of few-shot examples. We used zero-shot prompting for program and input-output test case generation, and few-shot prompting for the logical relations specifications generation. We drew samples from these models using nucleus sampling with temperature $= 0.8$ and TopP $= 0.95$.

**Logistic regressor.** We used the lbfgs solver from scitkit-learn We used repeated 5-fold nested cross-validation with regularization hyperparameter $C = 1$.

**Reproduciblity statement**

- Data: The HumanEval and MBPP dataset are from existing literature and are available through Azure cloud service. We will also release the samples to facilitate reproducibility.
- Code: We detailed our hyperparameter and also we will make the code public upon publication.

## B  DATASET STATISTICS

Below we show representative dataset statistics for Davinci Codex with temperature 0.8 and topP=0.95.

| | Input-Output | | Logical Relations | |
|---|---|---|---|---|
| | HumanEval | MBPP | HumanEval | MBPP |
| cluster size (# of programs) | 12.91 | 13.62 | 12.60 | 14.90 |
| stddev | 18.02 | 20.52 | 17.82 | 21.38 |
| average # of test cases per program | 348.99 | 334.67 | 100 | 100 |
| stddev | 137.18 | 146.47 | 0 | 0 |
| % of programs that satisfy at least one test | 84.02% | 82.49% | 88.73% | 82.43% |

## C  USER STUDY

In the framework of an Institutional Review Board (IRB)-approved study, we recruited 22 users, primarily consisting of Computer Science students. They were all compensated at rates exceeding the minimum wage, ensuring fair remuneration.

For the study, each participant was presented with a set of eight programs. These programs were samples from code-davinci-002 and consider top-rank programs by our system. Alongside each program, we offered three specifications: the top-ranked spec, the bottom-ranked spec, and a spec chosen at random. When the random-rank spec has the same score as the top-ranked or bottom-ranked spec, we re-sampled again.

Participants were requested to rate the utility of each specification using a 1-7 Likert scale. The scoring was based on their judgment of how effectively each spec assisted in determining the correctness or incorrectness of the corresponding program.

The results, illustrated in Fig. 7, establish that our heuristic is better than just the LLM on its own for generating interpretable explanations about program behavior that allow humans to decide whether a program is correct or not. The median values, around 5 for our approach, indicate a preference for the top-ranked specification over both the random and bottom ones. Besides, in the first quantile, our approach outperforms the random specification. In the third quantile, our heuristic achieves ratings of 6, surpassing the bottom-ranked specification.

In Figure 9, we provide a screenshot of the survey used for this study.

## D  PRECISION-RECALL CURVES

Figure 10 presents four distinct precision-recall curves. Each of these curves corresponds to one of the four different settings delineated in the table, as shown in Figure 3.

The illustrated curves provide a clear understanding of the relationship between precision and recall under various conditions. Interestingly, they demonstrate that high levels of precision can be achieved without substantially compromising the recall. This insight underscores the potential effectiveness of our approach in maintaining a balance between accurately identifying correct programs (precision) and still successfully retrieving correct programs in most cases(recall).

## E  GENERALIZATION ACROSS DATASETS

Unlike recent heuristics for reranking solutions proposed by a large language model, our scheme involves learning real-valued parameters ($\theta$ in Eq. (1). To understand how learned parameters generalize across datasets, we compute the *pass@1* rate and precision-recall stats for models trained on MBPP, but tested on HumanEval (and vice versa). These statistics are similar by training on different datasets (Fig. 11), indicating generalization across similar, but not identical, data distributions.

## F  EXAMPLE ZERO-SHOT PROMPTS FOR PROGRAM GENERATION

For MBPP, to generate programs, we converted the natural language prompt to a function by adding in the prompt as a docstring for a function with the name of the function called in the ground-truth test cases. We used the HumanEval prompts as is.

Two examples of zero-shot prompts used for program generation are as follows:

### F.1  HUMANEVAL

First example:

```
def is_happy(s):
    """You are given a string s.
    Your task is to check if the string is happy or not.
    A string is happy if its length is at least 3 and every 3
    ↪  consecutive letters are distinct
    For example:
    is_happy(a) => False
    is_happy(aa) => False
```

**Question 1:**

Please read the program below and **judge its implementation based on the docstring of the program**. Evaluate the usefulness of the following test cases in identifying whether the program is correctly or incorrectly implemented, using a scale of 1-7.

Remember that **the program passes all the test cases provided but the program can be either right or wrong** and your focus should be on the test cases' ability to help you determine the correctness or incorrectness of the program.

Possibly Correct Program

```python
def is_equal_to_sum_even(n):
    """Evaluate whether the given number n can be written as the sum of exactly 4 positive even numbers
    Example
    is_equal_to_sum_even(4) == False
    is_equal_to_sum_even(6) == False
    is_equal_to_sum_even(8) == True
    """
    return n % 2 == 0 and n >= 8
```

Please rate the following test *

```
assert is_equal_to_sum_even(13) == False
```

|  | 1 | 2 | 3 | 4 | 5 | 6 | 7 |  |
|---|---|---|---|---|---|---|---|---|
| Not Useful | ○ | ○ | ○ | ◉ | ○ | ○ | ○ | Very Useful |

Please rate the following test *

```
assert is_equal_to_sum_even(8589934592) == True
```

|  | 1 | 2 | 3 | 4 | 5 | 6 | 7 |  |
|---|---|---|---|---|---|---|---|---|
| Not Useful | ○ | ○ | ○ | ◉ | ○ | ○ | ○ | Very Useful |

Please rate the following test *

```
assert is_equal_to_sum_even(12) == True
```

|  | 1 | 2 | 3 | 4 | 5 | 6 | 7 |  |
|---|---|---|---|---|---|---|---|---|
| Not Useful | ○ | ○ | ○ | ◉ | ○ | ○ | ○ | Very Useful |

Figure 9: Screenshot of survey

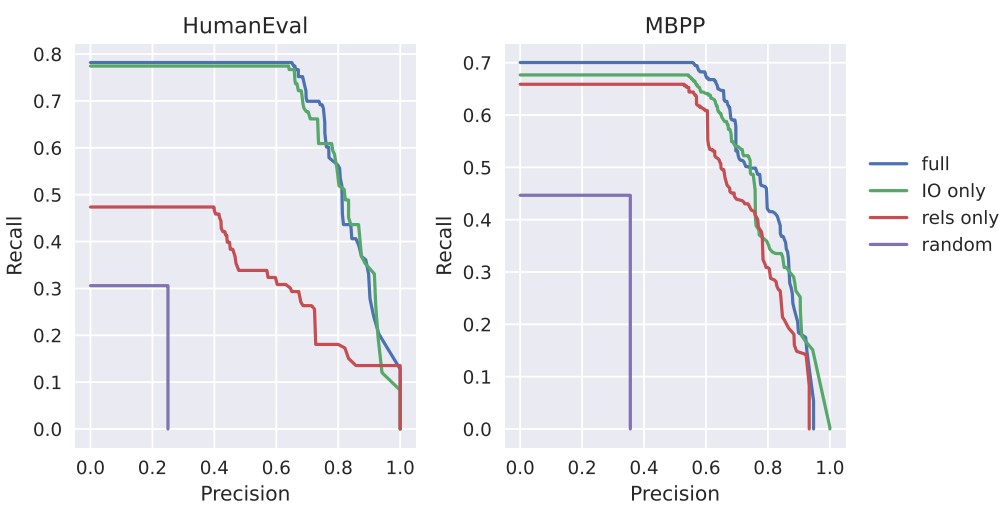

(a) Model: cushman-code-001

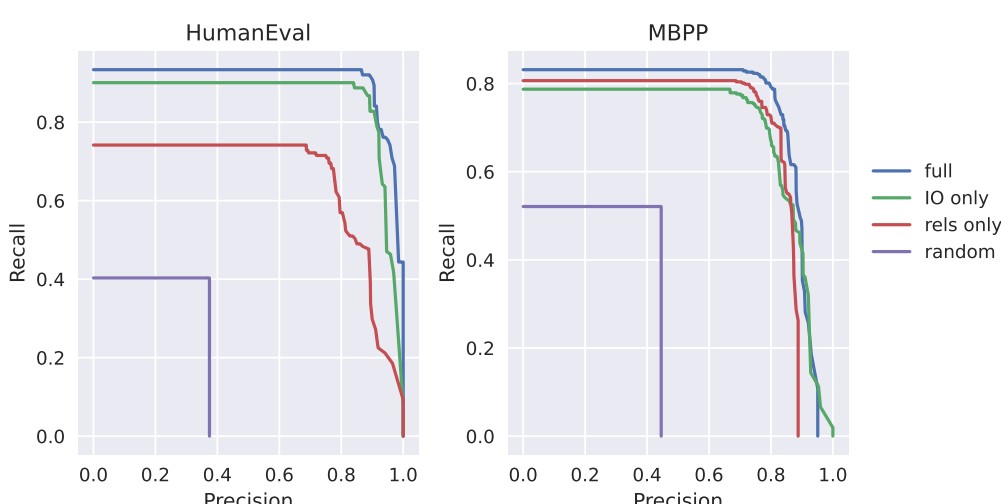

(b) Model: davinci-code-002

Figure 10: Precision-Recall curves for each model and dataset

| test | HumanEval | | MBPP | |
|---|---|---|---|---|
| train | HumanEval | MBPP | HumanEval | MBPP |
| pass@1 | 0.86 | 0.76 | 0.68 | 0.70 |
| AUC | 0.91 | 0.77 | 0.71 | 0.74 |
| max F1 | 0.91 | 0.81 | 0.76 | 0.80 |

Figure 11: Generalization when test/train data are drawn from the same corpus of problems, vs. drawn from different corpora.

```
is_happy(abcd) => True
is_happy(aabb) => False
is_happy(adb) => True
is_happy(xyy) => False
"""
```

Second example:

```
def fix_spaces(text):
    """
    Given a string text, replace all spaces in it with
    ↪  underscores,
    and if a string has more than 2 consecutive spaces,
    then replace all consecutive spaces with -

    fix_spaces("Example") == "Example"
    fix_spaces("Example 1") == "Example_1"
    fix_spaces(" Example 2") == "_Example_2"
    fix_spaces(" Example   3") == "_Example-3"
    """
```

## F.2 MBPP

First example:

```
def sum_range_list(list1 : list, m : int, n : int) -> int:
    """
    Write a function to find the sum of numbers in a list within
    ↪  a range specified by two indices.
    """
```

Second example:

```
def diff_even_odd(list1 : list) -> int:
    """
    Write a function to find the difference of the first even and
    ↪  first odd number of a given list.
    """
```

## G EXAMPLE ZERO-SHOT PROMPTS FOR INPUT-OUTPUT GENERATION

We extracted input-output test cases by generating $n = 100$ times per HumanEval/MBPP prompt, then extracting each distinct single-line test case from each generation. We do this because each generation may produce multiple test cases, and we aimed to test each program on a single test case. For our test case prompts, we used the prompts to generate programs from MBPP and HumanEval, and we added in a `pass # To-do:  implement` statement, a line with a comment asking Codex to `# Check if func_name works` and another line to asking Codex to `assert func_name(`.

### G.1 HUMANEVAL

First example:

```
def is_happy(s):
    """You are given a string s.
    Your task is to check if the string is happy or not.
    A string is happy if its length is at least 3 and every 3
    ↪  consecutive letters are distinct
    For example:
    is_happy(a) => False
    is_happy(aa) => False
    is_happy(abcd) => True
    is_happy(aabb) => False
    is_happy(adb) => True
    is_happy(xyy) => False
    """

    pass # To-do: implement

# Check if is_happy works
assert is_happy(
```

Second example:

```
def fix_spaces(text):
    """
    Given a string text, replace all spaces in it with
    ↪  underscores,
    and if a string has more than 2 consecutive spaces,
    then replace all consecutive spaces with -

    fix_spaces("Example") == "Example"
    fix_spaces("Example 1") == "Example_1"
    fix_spaces(" Example 2") == "_Example_2"
    fix_spaces(" Example   3") == "_Example-3"
    """

    pass # To-do: implement

# Check if fix_spaces works
assert fix_spaces(
```

### G.2 MBPP

First example:

```
def sum_range_list(list1 : list, m : int, n : int) -> int:
    """
    Write a function to find the sum of numbers in a list within
    ↪  a range specified by two indices.
    """
    pass # To-do: implement

# Check if sum_range_list works
assert sum_range_list(
```

Second example:

```
def diff_even_odd(list1 : list) -> int:
    """
    Write a function to find the difference of the first even and
    ↪  first odd number of a given list.
    """
    pass # To-do: implement

# Check if diff_even_odd works
assert diff_even_odd(
```

## H    FEW-SHOT PROMPT FOR LOGICAL RELATIONS SPEC GENERATION

We use two-shot and five-shot examples prompting to guide the model to tests various kinds of properties for 2k-context-size Cushman model and 8k-context-size Davinci model respectively.

### H.1    HUMANEVAL (CUSHMAN)

```
# Problem 1

from typing import List

def filtered_even_integers(input_list: List[int]) -> List[int]:
    """ Given a list of integers, return a list that filters out
    ↪  the even integers.
    >>>  filtered_even_integers([1, 2, 3, 4])
    [1, 3]
    >>> filtered_even_integers([5, 4, 3, 2, 1])
    [5, 3, 1]
    >>> filtered_even_integers([10, 18, 20])
    []
    """
    # TODO
    pass

# Test 1

def test_filtered_even_integers(input_list: List()):
    """ Given an input `input_list`, test whether the function
    ↪  `filtered_even_integers` is implemented correctly.
    """
    # execute the function
    output_list = filtered_even_integers(input_list)

    # check if the output list only contains odd integers
    for integer in output_list:
        assert integer % 2 == 1
    # check if all the integers in the output list can be found
    ↪  in the input list
    for integer in output_list:
        assert integer in input_list

# run the testing function `test_filtered_even_integers` on 3
↪  different input cases that satisfy the description
test_filtered_even_integers([31, 24, 18, 99, 1000, 523, 901])
test_filtered_even_integers([2, 4, 6, 8])
test_filtered_even_integers([500, 0, 302, 19, 7, 5])

# Problem 2
```

```
972
973  def repeat_vowel(input_str: str) -> str:
974      """ Return a string where the vowels (`a`, `e`, `i`, `o`, `u`,
975      ↪  and their capital letters) are repeated twice in place.
976      >>> repeat_vowel('abcdefg')
977      'aabcdeefg'
978      >>> repeat_vowel('Amy Emily Uber')
979      'AAmy EEmiily UUbeer'
980      """
981      # TODO
982      pass
983
984  # Test 2
985
986  def test_repeat_vowel(input_str: str) :
987      """ Given an input `input_str`, test whether the function
988      ↪  `repeat_vowel` is implemented correctly.
989      """
990      # execute the function
991      output_str = repeat_vowel(input_str)
992
993      vowels = ['a', 'A', 'e', 'E', 'i', 'I', 'o', 'O', 'u', 'U']
994      # check if the number of vowels in the  output string is
995      ↪  doubled
996      # First get the number of vowels in the input
997      number_of_vowels_input = sum([input_str.count(vowel) for
998      ↪  vowel in vowels])
999      # Then get the number of vowels in the output
1000     number_of_vowels_output = sum([output_str.count(vowel) for
1001     ↪  vowel in vowels])
1002     assert number_of_vowels_input * 2 == number_of_vowels_output
1003
1004 # run the testing function `test_repeat_vowel` on 3 different
1005 ↪  input cases that satisfy the description
1006 test_repeat_vowel('ABCDEabcdeABCDE YOUUOY')
1007 test_repeat_vowel('I am a student')
1008 test_repeat_vowel('sounds good to me')
```

## H.2  HUMANEVAL (DAVINCI)

```
# Problem 1

from typing import List

def filtered_even_integers(input_list: List[int]) -> List[int]:
    """ Given a list of integers, return a list that filters out
    ↪  the even integers.
    >>>  filtered_even_integers([1, 2, 3, 4])
    [1, 3]
    >>> filtered_even_integers([5, 4, 3, 2, 1])
    [5, 3, 1]
    >>> filtered_even_integers([10, 18, 20])
    []
    """
    # TODO
    pass

# Test 1
```

```
1026   def test_filtered_even_integers(input_list: List()):
1027       """ Given an input `input_list`, test whether the function
1028       ↪  `filtered_even_integers` is implemented correctly.
1029       """
1030       # execute the function
1031       output_list = filtered_even_integers(input_list)
1032
1033       # check if the output list only contains odd integers
1034       for integer in output_list:
1035           assert integer % 2 == 1
1036       # check if all the integers in the output list can be found
1036       ↪  in the input list
1037       for integer in output_list:
1038           assert integer in input_list
1039
1040   # run the testing function `test_filtered_even_integers` on 3
1041   ↪  different input cases that satisfy the description
1042   test_filtered_even_integers([31, 24, 18, 99, 1000, 523, 901])
1043   test_filtered_even_integers([2, 4, 6, 8])
1044   test_filtered_even_integers([500, 0, 302, 19, 7, 5])
1045
1046   # Problem 2
1047
1048   def repeat_vowel(input_str: str) -> str:
1049       """ Return a string where the vowels (`a`, `e`, `i`, `o`, `u`,
1049       ↪  and their capital letters) are repeated twice in place.
1050       >>> repeat_vowel('abcdefg')
1051       'aabcdeefg'
1052       >>> repeat_vowel('Amy Emily Uber')
1053       'AAmy EEmiily UUbeer'
1054       """
1055       # TODO
1056       pass
1057
1058   # Test 2
1059
1060   def test_repeat_vowel(input_str: str) :
1061       """ Given an input `input_str`, test whether the function
1062       ↪  `repeat_vowel` is implemented correctly.
1062       """
1063       # execute the function
1064       output_str = repeat_vowel(input_str)
1065
1066       vowels = ['a', 'A', 'e', 'E', 'i', 'I', 'o', 'O', 'u', 'U']
1067       # check if the number of vowels in the  output string is
1068       ↪  doubled
1069       # First get the number of vowels in the input
1070       number_of_vowels_input = sum([input_str.count(vowel) for
1070       ↪  vowel in vowels])
1071       # Then get the number of vowels in the output
1072       number_of_vowels_output = sum([output_str.count(vowel) for
1073       ↪  vowel in vowels])
1074       assert number_of_vowels_input * 2 == number_of_vowels_output
1075
1076   # run the testing function `test_repeat_vowel` on 3 different
1077   ↪  input cases that satisfy the description
1078   test_repeat_vowel('ABCDEabcdeABCDE YOUUOY')
1079   test_repeat_vowel('I am a student')
       test_repeat_vowel('sounds good to me')
```

```
1080
1081    # Problem 3
1082
1083    def find_missing_number(nums: List[int]) -> int:
1084        """
1085        Given a list of n-1 integers in the range of 1 to n, find the
1086        ↪  one missing number.
1087        >>> find_missing_number([1, 2, 4, 6, 3, 7, 8])
1088        5
1089        >>> find_missing_number([5, 1, 4, 2])
1090        3
1091        """
1092        # TODO
1093        pass
1094
1095    # Test 3
1096
1097    def test_find_missing_number(nums: List[int]):
1098        """ Given an input `nums`, test whether the function
1099        ↪  `find_missing_number` is implemented correctly.
1100        """
1101
1102        # execute the function
1103        output = find_missing_number(nums)
1104
1105        n = len(nums) + 1
1106        # check if the output is an integer
1107        assert isinstance(output, int)
1108        # check if the output is in the range of 1 to n
1109        assert 1 <= output <= n
1110        # check if the output is the missing number
1111        assert output not in nums
1112
1113    # run the testing function `test_find_missing_number` on 3
1114    ↪  different input cases that satisfy the description
1115    test_find_missing_number([1, 3, 4, 6, 5, 7, 8])
1116    test_find_missing_number([10, 9, 8, 7, 6, 5, 4, 3, 2])
1117    test_find_missing_number([3, 2, 1, 6, 5, 4, 10, 9, 8])
1118
1119    # Problem 4
1120
1121    def find_kth_largest(nums: List[int], k: int) -> int:
1122        """
1123        Given an unsorted array of integers, find the kth largest
1124        ↪  element.
1125        >>> find_kth_largest([3, 2, 1, 5, 6, 4], 2)
1126        5
1127        """
1128        # TODO
1129        pass
1130
1131    # Test 4
1132
1133    def test_find_kth_largest(nums: List[int], k: int):
        """ Given an input `nums` and `k`, test whether the function
        ↪  `find_kth_largest` is implemented correctly.
        """
        # execute the function
        output = find_kth_largest(nums, k)
```

```
1134
1135        # check if the output is an integer
1136        assert isinstance(output, int)
1137        # check if the output is in the input list
1138        assert output in nums
1139        # check if the output is the kth largest element
1140        assert output == sorted(nums)[-k]
1141
1142    # run the testing function `test_find_kth_largest` on 3 different
1143    ↪   input cases that satisfy the description
1144    test_find_kth_largest([1, 10, 9, 2, 7, 6, -3, 4, 5, 8], 3)
1145    test_find_kth_largest([100, 200, 900, 1000, 80, 101010], 5)
1146    test_find_kth_largest([88, 131, 89, 125, 3, 7], 2)
1147
1148    # Problem 5
1149
1150    def reverse_substrings(s: str, indices: List[int]) -> str:
1151        """
1152        Given a string s and a list of integers representing starting
1153        ↪   and ending indices of substrings
1154        within s (inclusive), reverse each substring and return the
1155        ↪   modified string.
1156        >>> reverse_substrings('abcdefg', [1, 2, 4, 6])
1157        'acbdgfe'
1158        """
1159        # TODO
1160        pass
1161
1162    # Test 5
1163
1164    def test_reverse_substrings(s: str, indices: List[int]):
1165        """ Given an input `s` and `indices`, test whether the
1166        ↪   function `reverse_substrings` is implemented correctly.
1167        """
1168        # execute the function
1169        output = reverse_substrings(s, indices)
1170
1171        # check if the function is implemented correctly
1172        # check if the output is a string
1173        assert isinstance(output, str)
1174        # check if the output is the same length as the input
1175        assert len(output) == len(s)
1176        # check if the output contains the same characters as the
1177        ↪   input
1178        assert set(output) == set(s)
1179        # check if all the substrings are reversed
1180        for i in range(0, len(indices), 2):
1181            assert output[indices[i]:indices[i+1]+1] ==
1182            ↪   s[indices[i]:indices[i+1]+1][::-1]
1183        # check if all the other characters are the same
1184        for i in range(len(s)):
1185            # check if i in the indices
1186            if any(indices[i] <= i <= indices[i+1] for i in range(0,
1187            ↪   len(indices), 2)):
                    continue
                assert output[i] == s[i]

    # run the testing function `test_reverse_substrings` on 3
    ↪   different input cases that satisfy the description
```

```
1188  test_reverse_substrings('apple', [1, 2, 4, 5])
1189  test_reverse_substrings('summerSpringWinterfall', [0, 5, 8, 9,
1190  ↪   10, 15, 16, 20])
1191  test_reverse_substrings('lkjhgfdqwert', [0, 3, 4, 7, 9, 11])
1192
```

### H.3   MBPP (CUSHMAN)

```
1195  # Problem 1
1196
1197  from typing import List
1198
1199  def filtered_even_integers(input_list: List[int]) -> List[int]:
1200      """
1201      Given a list of integers, return a list that filters out the
1202      ↪   even integers.
1203      """
1204      # TODO
1205      pass
1206
1207  # Test 1
1208
1209  def test_filtered_even_integers(input_list: List()):
1210      """
1211      Given an input `input_list`, test whether the function
1212      ↪   `filtered_even_integers` is implemented correctly.
1213      """
1214      # execute the function
1215      output_list = filtered_even_integers(input_list)
1216
1217      # check if the output list only contains odd integers
1218      for integer in output_list:
1219          assert integer % 2 == 1
1220      # check if all the integers in the output list can be found
1221      ↪   in the input list
1222      for integer in output_list:
1223          assert integer in input_list
1224
1225  # run the testing function `test_filtered_even_integers` on 3
1226  ↪   different input cases that satisfy the description
1227  test_filtered_even_integers([31, 24, 18, 99, 1000, 523, 901])
1228  test_filtered_even_integers([2, 4, 6, 8])
1229  test_filtered_even_integers([500, 0, 302, 19, 7, 5])
1230
1231  # Problem 2
1232
1233  def repeat_vowel(input_str: str) -> str:
1234      """
1235      Return a string where the vowels (`a`, `e`, `i`, `o`, `u`, and
1236      ↪   their capital letters) are repeated twice in place.
1237      """
1238      # TODO
1239      pass
1240
1241  # Test 2

      def test_repeat_vowel(input_str: str) :
          """
          Given an input `input_str`, test whether the function
          ↪   `repeat_vowel` is implemented correctly.
```

```
1242        """
1243        # execute the function
1244        output_str = repeat_vowel(input_str)
1245
1246        vowels = ['a', 'A', 'e', 'E', 'i', 'I', 'o', 'O', 'u', 'U']
1247        # check if the number of vowels in the  output string is
1248        ↪   doubled
1249        # First get the number of vowels in the input
1250        number_of_vowels_input = sum([input_str.count(vowel) for
1251        ↪   vowel in vowels])
1252        # Then get the number of vowels in the output
1253        number_of_vowels_output = sum([output_str.count(vowel) for
1254        ↪   vowel in vowels])
1255        assert number_of_vowels_input * 2 == number_of_vowels_output

1256   # run the testing function `test_repeat_vowel` on 3 different
1257   ↪   input cases that satisfy the description
1258   test_repeat_vowel('ABCDEabcdeABCDE YOUUOY')
1259   test_repeat_vowel('I am a student')
1260   test_repeat_vowel('sounds good to me')
```

## H.4 MBPP (DAVINCI)

```
# Problem 1

from typing import List

def filtered_even_integers(input_list: List[int]) -> List[int]:
    """
    Given a list of integers, return a list that filters out the
    ↪   even integers.
    """
    # TODO
    pass

# Test 1

def test_filtered_even_integers(input_list: List()):
    """
    Given an input `input_list`, test whether the function
    ↪   `filtered_even_integers` is implemented correctly.
    """
    # execute the function
    output_list = filtered_even_integers(input_list)

    # check if the output list only contains odd integers
    for integer in output_list:
        assert integer % 2 == 1
    # check if all the integers in the output list can be found
    ↪   in the input list
    for integer in output_list:
        assert integer in input_list

# run the testing function `test_filtered_even_integers` on 3
↪   different input cases that satisfy the description
test_filtered_even_integers([31, 24, 18, 99, 1000, 523, 901])
test_filtered_even_integers([2, 4, 6, 8])
test_filtered_even_integers([500, 0, 302, 19, 7, 5])
```

```
# Problem 2

def repeat_vowel(input_str: str) -> str:
    """
    Return a string where the vowels (`a`, `e`, `i`, `o`, `u`, and
    ↪  their capital letters) are repeated twice in place.
    """
    # TODO
    pass

# Test 2

def test_repeat_vowel(input_str: str) :
    """
    Given an input `input_str`, test whether the function
    ↪  `repeat_vowel` is implemented correctly.
    """
    # execute the function
    output_str = repeat_vowel(input_str)

    vowels = ['a', 'A', 'e', 'E', 'i', 'I', 'o', 'O', 'u', 'U']
    # check if the number of vowels in the  output string is
    ↪  doubled
    # First get the number of vowels in the input
    number_of_vowels_input = sum([input_str.count(vowel) for
    ↪  vowel in vowels])
    # Then get the number of vowels in the output
    number_of_vowels_output = sum([output_str.count(vowel) for
    ↪  vowel in vowels])
    assert number_of_vowels_input * 2 == number_of_vowels_output

# run the testing function `test_repeat_vowel` on 3 different
↪  input cases that satisfy the description
test_repeat_vowel('ABCDEabcdeABCDE YOUUOY')
test_repeat_vowel('I am a student')
test_repeat_vowel('sounds good to me')

# Problem 3

def find_missing_number(nums: List[int]) -> int:
    """
    Given a list of n-1 integers in the range of 1 to n, find the
    ↪  one missing number.
    """
    # TODO
    pass

# Test 3

def test_find_missing_number(nums: List[int]):
    """
    Given an input `nums`, test whether the function
    ↪  `find_missing_number` is implemented correctly.
    """

    # execute the function
    output = find_missing_number(nums)

    n = len(nums) + 1
```

```
        # check if the output is an integer
        assert isinstance(output, int)
        # check if the output is in the range of 1 to n
        assert 1 <= output <= n
        # check if the output is the missing number
        assert output not in nums

# run the testing function `test_find_missing_number` on 3
↪  different input cases that satisfy the description
test_find_missing_number([1, 3, 4, 6, 5, 7, 8])
test_find_missing_number([10, 9, 8, 7, 6, 5, 4, 3, 2])
test_find_missing_number([3, 2, 1, 6, 5, 4, 10, 9, 8])

# Problem 4

def find_kth_largest(nums: List[int], k: int) -> int:
    """
    Given an unsorted array of integers, find the kth largest
    ↪  element.
    """
    # TODO
    pass

# Test 4

def test_find_kth_largest(nums: List[int], k: int):
    """
    Given an input `nums` and `k`, test whether the function
    ↪  `find_kth_largest` is implemented correctly.
    """
    # execute the function
    output = find_kth_largest(nums, k)

    # check if the output is an integer
    assert isinstance(output, int)
    # check if the output is in the input list
    assert output in nums
    # check if the output is the kth largest element
    assert output == sorted(nums)[-k]

# run the testing function `test_find_kth_largest` on 3 different
↪  input cases that satisfy the description
test_find_kth_largest([1, 10, 9, 2, 7, 6, -3, 4, 5, 8], 3)
test_find_kth_largest([100, 200, 900, 1000, 80, 101010], 5)
test_find_kth_largest([88, 131, 89, 125, 3, 7], 2)

# Problem 5

def reverse_substrings(s: str, indices: List[int]) -> str:
    """
    Given a string s and a list of integers representing starting
    ↪  and ending indices of substrings
    within s (inclusive), reverse each substring and return the
    ↪  modified string.
    """
    # TODO
    pass

# Test 5
```

```
def test_reverse_substrings(s: str, indices: List[int]):
    """
    Given an input `s` and `indices`, test whether the function
    ↪  `reverse_substrings` is implemented correctly.
    """
    # execute the function
    output = reverse_substrings(s, indices)

    # check if the function is implemented correctly
    # check if the output is a string
    assert isinstance(output, str)
    # check if the output is the same length as the input
    assert len(output) == len(s)
    # check if the output contains the same characters as the
    ↪  input
    assert set(output) == set(s)
    # check if all the substrings are reversed
    for i in range(0, len(indices), 2):
        assert output[indices[i]:indices[i+1]+1] ==
        ↪  s[indices[i]:indices[i+1]+1][::-1]
    # check if all the other characters are the same
    for i in range(len(s)):
        # check if i in the indices
        if any(indices[i] <= i <= indices[i+1] for i in range(0,
        ↪  len(indices), 2)):
            continue
        assert output[i] == s[i]

# run the testing function `test_reverse_substrings` on 3
↪  different input cases that satisfy the description
test_reverse_substrings('apple', [1, 2, 4, 5])
test_reverse_substrings('summerSpringWinterfall', [0, 5, 8, 9,
↪  10, 15, 16, 20])
test_reverse_substrings('lkjhgfdqwert', [0, 3, 4, 7, 9, 11])
```

# I  TRANSFORMATION OF INPUT PROBLEMS TO LOGICAL RELATIONS PROMPTS

Here we show how to transform the input problem to the prompt used for generating logical relations.

**MBPP transformation**  First we parse the input problems from MBPP dataset and get the string representation of library imports, function name, function parameters, return type, and English problem description. We denote them as `imports`, `func_name`, `parameter_format`, `return_type`, and `description` respectively and `problem_number` is number of few-shot examples plus 1.

Then we use the template shown in Figure 13 and Figure 12 for input-output and logical relations respectively. The parsed string from the input problem would then be inserted to the placeholder accordingly.

```
# Problem 3

def {func_name}({", ".join(parameter_format)}) -> {return_type}:
    """
    {description}
    """
    pass # To-do: implement

# Test 3

def test_{func_name}(
```

Figure 12: Template for MBPP logical relation prompt

```
# Problem {problem_number}

{function_definition_with_description}
    # TODO
    pass

# Test {problem_number}
```

Figure 13: Template for MBPP input-output prompt

Then, for the logical relations, we prepend the resulting string with the few shot example string shown in H.3. For the input-output, we strip out the blank lines prefix if import is empty.

**HumanEval Transformation**  Similar to the above MBPP transformation, we parse the input problems from HumanEval dataset and get the string representation of function definition plus English description and function name.  We denote them as function_definition_with_description and func_name.  Then we insert these into the template shown in Figure 14 and Figure 15 for input-output and logical relations respectively.

```
{function_definition_with_description}
    pass

# Check if {func_name} works
assert {func_name}(
```

Figure 14: Template for HumanEval input-output prompt

```
# Problem {problem_number}

{function_definition_with_description}
    # TODO
    pass

# Test {problem_number}
```

Figure 15: Template for HumanEval logical relation prompt

Finally, we prepend the resulting string with the few shot example string shown in H.1.

```python
def compare_one(a, b):
    """
    Create a function that takes integers, floats, or strings representing
    real numbers, and returns the larger variable in its given variable type.
    Return None if the values are equal.
    Note: If a real number is represented as a string, the floating point might be . or ,

    compare_one(1, 2.5) ➞ 2.5
    compare_one(1, "2,3") ➞ "2,3"
    compare_one("5,1", "6") ➞ "6"
    compare_one("1", 1) ➞ None
    """
    if type(a) == type(b):
        try:
            if float(a) > float(b):
                return a
            elif float(a) < float(b):
                return b
            else:
                return None
        except:
            if str(a) > str(b):
                return a
            elif str(a) < str(b):
                return b
            else:
                return None
    else:
        try:
            if float(a) > float(b):
                return a
            elif float(a) < float(b):
                return b
            else:
                return None
        except:
            try:
                if float(b) > float(a):
                    return b
                elif float(b) < float(a):
                    return a
                else:
                    return None
            except:
                if str(a) > str(b):
                    return a
                elif str(a) < str(b):
                    return b
                else:
                    return None
```

(a) Program 1

## J    EXAMPLES OF TOP SPEC VERSUS RANDOM SPEC

In the following pages, we introduce 8 example HumanEval problems along with program samples generated by the code-davinci-002 model. For each of these programs, we display our top-ranked specification, a randomly chosen specification, and a bottom-ranked specification. Among them, four illustrate relation specifications, while the remaining present input/output (IO) specifications.

```python
def test_compare_one(a, b):
    """Given an input `a` and `b`, test whether the function `compare_one` is implemented correctly."""

    # execute the function
    output = compare_one(a, b)

    # check if the output is implemented correctly
    # check if the output is either a float, a string, an int or None
    assert isinstance(output, (float, str, int)) or output is None
    # check if the output is the larger of the two arguments if they are both real numbers
    if (
        isinstance(a, (float, int)) or (isinstance(a, str) and a.replace(",", ".").isdecimal())
    ) and (isinstance(b, (float, int)) or (isinstance(b, str) and b.replace(",", ".").isdecimal())):
        if float(a) > float(b):
            assert float(output) == float(a)
        if float(a) < float(b):
            assert float(output) == float(b)
        if float(a) == float(b):
            assert output is None

# run the testing function `test_compare_one` on 3 different input cases that satisfy the description
test_compare_one(1, 2.5)
test_compare_one(1, "2,3")
test_compare_one("5.1", "6")
test_compare_one("1", 1)
```

(b) Top-ranked Relation Spec for Program 1

```python
def test_compare_one(a, b):
    """Given an input `a` and `b`, test whether the function `compare_one` is implemented correctly."""
    # execute the function
    output = compare_one(a, b)

    # check if the output is the larger variable in its given variable type or None if the values are equal
    if a > b:
        assert output == a
    elif b > a:
        assert output == b
    else:
        assert output is None

# run the testing function `test_compare_one` on 3 different input cases that satisfy the description
test_compare_one(1, 3)
test_compare_one(1.1, 1)
test_compare_one(2, 2.2)
test_compare_one(1, 1)
```

(c) Random Relation Spec for Program 1

```python
def test_compare_one(a, b):
    # TODO: implement a test function for Problem 6
    # assert the result of compare_one(a, b) and compare_one(b, a) is the same
    pass

test_compare_one(1, 2.5)
test_compare_one(1, "2,3")
test_compare_one("5,1", "6")
test_compare_one("1", 1)
```

(d) Bottom-Ranked Relation Spec for Program 1

Figure 16: Examples of top-ranked, random, bottom-ranked specifications

```python
def digits(n):
    """Given a positive integer n, return the product of the odd digits.
    Return 0 if all digits are even.
    For example:
    digits(1)   == 1
    digits(4)   == 0
    digits(235) == 15
    """

    result = 1

    for x in str(n):
        if int(x) % 2 != 0:
            result *= int(x)
        else:
            result = 0

    return result
```

(a) Program 2

```python
assert digits(98989898) == 0
```

(b) Top-ranked IO Spec for Program 2

```python
assert digits(1) == 1
```

(c) Random IO Spec for Program 2

```python
assert digits(9) == 9
```

(d) Bottom-Ranked IO Spec for Program 2

Figure 17: Examples of top-ranked, random, bottom-ranked specifications for Program 2

```python
def file_name_check(file_name):
    """Create a function which takes a string representing a file's name, and returns
    'Yes' if the the file's name is valid, and returns 'No' otherwise.
    A file's name is considered to be valid if and only if all the following conditions
    are met:
    - There should not be more than three digits ('0'-'9') in the file's name.
    - The file's name contains exactly one dot '.'
    - The substring before the dot should not be empty, and it starts with a letter from
    the latin alphapet ('a'-'z' and 'A'-'Z').
    - The substring after the dot should be one of these: ['txt', 'exe', 'dll']
    Examples:
    file_name_check("example.txt") # => 'Yes'
    file_name_check("1example.dll") # => 'No' (the name should start with a latin alphapet letter)
    """

    def is_digit_valid(file_name):
        count = 0
        for c in file_name:
            if c.isdigit():
                count += 1
        return count < 4

    def is_dot_valid(file_name):
        count = 0
        for c in file_name:
            if c == ".":
                count += 1
        return count == 1

    def is_ext_valid(file_name):
        dot = file_name.rfind(".")
        ext = file_name[dot:]
        return ext in [".txt", ".exe", ".dll"]

    def is_name_valid(file_name):
        dot = file_name.rfind(".")
        name = file_name[:dot]
        if len(name) == 0:
            return False
        return name[0].isalpha()

    if (
        is_digit_valid(file_name)
        and is_dot_valid(file_name)
        and is_ext_valid(file_name)
        and is_name_valid(file_name)
    ):
        return "Yes"
    else:
        return "No"
```

(a) Program 3

```python
def test_file_name_check(file_name, expected):
    """
    Given an input `file_name`, test whether the function `file_name_check` is implemented correctly.
    """
    # execute the function
    output = file_name_check(file_name)
    # check if the function is implemented correctly
    assert output == expected

# run the testing function `test_file_name_check` on 6 different input cases that satisfy the description
test_file_name_check("example.txt", "Yes")
test_file_name_check("1example.dll", "No")
test_file_name_check("example.dll", "Yes")
test_file_name_check("example", "No")
test_file_name_check("example1112.txt", "No")
test_file_name_check("example1112.exe", "No")
```

(b) Top-ranked Relation Spec for Program 3

```python
def test_file_name_check(file_name):
    """Given an input `file_name`, test whether the function `file_name_check` is implemented correctly."""

    # execute the function
    output = file_name_check(file_name)

    # check if the function is implemented correctly
    if output == "Yes":
        # check if the string contains a dot
        assert len([1 for c in file_name if c == "."]) == 1
        # check if there are more than three digits in the name
        # get all the digits
        digits = [c for c in file_name if c in "0123456789"]
        assert len(digits) <= 3
        # get the substring before the dot
        before_dot = file_name.split(".")[0]
        # check if the substring is not empty
        assert before_dot
        # check if it starts with a latin alphapet letter
        assert before_dot[0] in "ABCDEFGHIJKLMNOPQRSTUVWXYZabcdefghijklmnopqrstuvwxyz"
        # get the substring after the dot
        after_dot = file_name.split(".")[1]
        # check if it is a valid extension
        assert after_dot in ["txt", "exe", "dll"]
    else:
        # check if the output is 'No'
        assert output == "No"

# run the testing function `test_file_name_check` on 3 different input cases that satisfy the description
test_file_name_check("example.txt")
test_file_name_check("1example.dll")
test_file_name_check("0123456789.txt")
```

(c) Random Relation Spec for Program 3

```python
def test_file_name_check(file_name: str):
    """Given an input `file_name`, test whether the function `file_name_check` is implemented correctly."""
    # execute the function
    output = file_name_check(file_name)

    # check if the function is implemented correctly
    # check if the output is a string
    assert isinstance(output, str)
    # check if the output is one of the two values 'Yes' or 'No'
    assert output in ["Yes", "No"]
    # if the output is 'No', then the function should return 'No'
    if output == "No":
        return
    # if the output is 'Yes', then there should not be more than three digits in the file name
    digits_count = sum(map(str.isdigit, file_name))
    assert digits_count <= 3
    # if the output is 'Yes', then the file name should contain exactly one dot
    assert sum(map(str.count, file_name, ["."] * len(file_name))) == 1
    # if the output is 'Yes', then the substring before the dot should not be empty, and start with a letter from latin alphabet
    name_before_dot = file_name.split(".")[0]
    assert len(name_before_dot) > 0 and name_before_dot[0] in [
        chr(x) for x in range(ord("a"), ord("z") + 1)
    ] + [chr(x) for x in range(ord("A"), ord("Z") + 1)]
    # if the output is 'Yes', then the substring after the dot should be one of the three values 'txt', 'exe' or 'dll'
    assert file_name.split(".")[1] in ["txt", "exe", "dll"]

# run the testing function `test_file_name_check` on 3 different input cases that satisfy the description
test_file_name_check("example.txt")
test_file_name_check("test.exe")
test_file_name_check("abc.dll")
```

(d) Bottom-Ranked Relation Spec for Program 3

Figure 18: Examples of top-ranked, random, bottom-ranked specifications for Program 3

```python
def is_equal_to_sum_even(n):
    """Evaluate whether the given number n can be written as the sum of exactly 4 positive even numbers
    Example
    is_equal_to_sum_even(4) == False
    is_equal_to_sum_even(6) == False
    is_equal_to_sum_even(8) == True
    """
    return n % 2 == 0 and n >= 8
```

(a) Program 4

```python
assert is_equal_to_sum_even(8589934592) == True
```

(b) Top-Ranked IO Spec for Program 4

```python
assert is_equal_to_sum_even(12) == True
```

(c) Random IO Spec for Program 4

```python
assert is_equal_to_sum_even(13) == False
```

(d) Bottom-Ranked IO Spec for Program 4

Figure 19: Examples of top-ranked, random, bottom-ranked specifications for Program 4

```python
def match_parens(lst):
    """
    You are given a list of two strings, both strings consist of open
    parentheses '(' or close parentheses ')' only.
    Your job is to check if it is possible to concatenate the two strings in
    some order, that the resulting string will be good.
    A string S is considered to be good if and only if all parentheses in S
    are balanced. For example: the string '(())()' is good, while the string
    '())' is not.
    Return 'Yes' if there's a way to make a good string, and return 'No' otherwise.

    Examples:
    match_parens(['()(', ')']) == 'Yes'
    match_parens([')', ')']) == 'No'
    """
    s = "".join(lst)

    # A stack data structure.
    stack = []

    # Check every parenthesis in the string.
    for char in s:
        if char == "(":
            # If it's an open parenthesis, push it to the stack.
            stack.append(char)
        elif char == ")":
            # If it's a close parenthesis.
            if len(stack) == 0:
                # If stack is empty, it means that there's no open parenthesis
                # to match this close parenthesis, so it's not a valid
                # string, return False.
                return "No"
            else:
                # If stack is not empty, we pop from the stack.
                stack.pop()
    # If stack is empty, it means the string is valid, return True.
    if len(stack) == 0:
        return "Yes"
    else:
        # If stack is not empty, it means there's open parentheses without
        # matching close parentheses, so it's not a valid string, return False.
        return "No"
```

(a) Program 5

```python
def test_match_parens(lst):
    """
    You are given a list of two strings, both strings consist of open
    parentheses '(' or close parentheses ')' only.
    Your job is to check if it is possible to concatenate the two strings in
    some order, that the resulting string will be good.
    A string S is considered to be good if and only if all parentheses in S
    are balanced. For example: the string '(())()' is good, while the string
    '())' is not.
    Return 'Yes' if there's a way to make a good string, and return 'No' otherwise.

    Examples:
    match_parens(['()(', ')']) == 'Yes'
    match_parens([')', ')']) == 'No'
    """
    # execute the function
    output = match_parens(lst)

    # check if the output is a string
    assert isinstance(output, str)
    # check if the output is 'Yes' or 'No'
    assert output in ["Yes", "No"]
    # check if the output satisfies the description
    if output == "Yes":
        assert _is_valid_parens("".join(lst))
    if output == "No":
        assert not _is_valid_parens("".join(lst))

def _is_valid_parens(s):
    """
    Return whether the string of parentheses is a valid one.
    """
    stack = []
    for char in s:
        if char == "(":
            stack.append("(")
        elif stack:
            stack.pop()
        else:
            return False
    return not stack

# run the testing function `test_match_parens` on 3 different input cases that satisfy the description
test_match_parens(["()", "(", ")"])
test_match_parens(["(", ")", "("])
test_match_parens(["(", "(", ")"])
```

(b) Top-ranked Relation Spec for Program 5

```python
def test_match_parens(lst):
    """
    Given an input `lst`, test whether the function `match_parens` is implemented correctly.
    """
    # execute the function
    output = match_parens(lst)

    # check if the output is a string
    assert isinstance(output, str)

    # check if the output is 'Yes' or 'No'
    assert output in ["Yes", "No"]

    # check if the output is correct
    if output == "Yes":
        s = "".join(lst)
        # check if the parentheses are matched
        # TODO
        pass
    elif output == "No":
        # TODO
        pass

# run the testing function `test_match_parens` on 3 different input cases that satisfy the description
test_match_parens(["(", ")", "(", ")", "()", "((()))"])
test_match_parens(["((", "))"])
test_match_parens(["(", ")", ")", ")", "("])
```

(c) Random Relation Spec for Program 5

```python
def test_match_parens(lst):
    """
    Given a list lst, test whether the function match_parens is implemented correctly
    """
    # TODO
    pass

# run the testing function test_match_parens on 3 different input cases that satisfy the description
test_match_parens(["()(", ")"])
test_match_parens([")", ")"])
test_match_parens(["(((((((", ")))()()"])
```

(d) Bottom-Ranked Relation Spec for Program 5

Figure 20: Examples of top-ranked, random, bottom-ranked specifications for Program 5

```python
def minPath(grid, k):
    """
    Given a grid with N rows and N columns (N >= 2) and a positive integer k,
    each cell of the grid contains a value. Every integer in the range [1, N * N]
    inclusive appears exactly once on the cells of the grid.

    You have to find the minimum path of length k in the grid. You can start
    from any cell, and in each step you can move to any of the neighbor cells,
    in other words, you can go to cells which share an edge with you current
    cell.
    Please note that a path of length k means visiting exactly k cells (not
    necessarily distinct).
    You CANNOT go off the grid.
    A path A (of length k) is considered less than a path B (of length k) if
    after making the ordered lists of the values on the cells that A and B go
    through (let's call them lst_A and lst_B), lst_A is lexicographically less
    than lst_B, in other words, there exist an integer index i (1 <= i <= k)
    such that lst_A[i] < lst_B[i] and for any j (1 <= j < i) we have
    lst_A[j] = lst_B[j].
    It is guaranteed that the answer is unique.
    Return an ordered list of the values on the cells that the minimum path go through.

    Examples:

        Input: grid = [ [1,2,3], [4,5,6], [7,8,9]], k = 3
        Output: [1, 2, 1]

        Input: grid = [ [5,9,3], [4,1,6], [7,8,2]], k = 1
        Output: [1]
    """
    n = len(grid)
    visited = [[False] * n for _ in range(n)]
    res = None

    def dfs(x, y, k, path):
        if k == 0:
            nonlocal res
            if not res or path < res:
                res = path
            return
        else:
            for nx, ny in (x - 1, y), (x + 1, y), (x, y - 1), (x, y + 1):
                if 0 <= nx < n and 0 <= ny < n and not visited[nx][ny]:
                    visited[nx][ny] = True
                    dfs(nx, ny, k - 1, path + [grid[nx][ny]])
                    visited[nx][ny] = False

    for i in range(n):
        for j in range(n):
            visited[i][j] = True
            dfs(i, j, k - 1, [grid[i][j]])
            visited[i][j] = False

    return res
```

(a) Program 6

```
assert minPath([[5, 9, 3], [4, 1, 6], [7, 8, 2]], 2) == [1, 4]
```

(b) Top-ranked IO Spec for Program 6

```
assert minPath(grid=[[5, 9, 3], [4, 1, 6], [7, 8, 2]], k=1) == [1]
```

(c) Random Spec IO for Program 6

```
assert minPath([[9, 9, 9], [9, 9, 9], [9, 9, 9]], 1) == [9]
```

(d) Bottom-Ranked IO Spec for Program 6

Figure 21: Examples of top-ranked, random, bottom-ranked specifications for Program 6

```python
def move_one_ball(arr):
    """We have an array 'arr' of N integers arr[1], arr[2], ..., arr[N].The
    numbers in the array will be randomly ordered. Your task is to determine if
    it is possible to get an array sorted in non-decreasing order by performing
    the following operation on the given array:
        You are allowed to perform right shift operation any number of times.

    One right shift operation means shifting all elements of the array by one
    position in the right direction. The last element of the array will be moved to
    the starting position in the array i.e. 0th index.

    If it is possible to obtain the sorted array by performing the above operation
    then return True else return False.
    If the given array is empty then return True.

    Note: The given list is guaranteed to have unique elements.

    For Example:

    move_one_ball([3, 4, 5, 1, 2])==>True
    Explanation: By performin 2 right shift operations, non-decreasing order can
                 be achieved for the given array.
    move_one_ball([3, 5, 4, 1, 2])==>False
    Explanation:It is not possible to get non-decreasing order for the given
                array by performing any number of right shift operations.

    """
    if len(arr) == 0:
        return True

    for i in range(len(arr)):
        for j in range(len(arr) - 1):
            if arr[j] > arr[j + 1]:
                temp = arr[j]
                arr[j] = arr[j + 1]
                arr[j + 1] = temp

    return arr == sorted(arr)
```

(a) Program 7

```python
def test_move_one_ball(arr):
    """Given an input `arr`, test whether the function `move_one_ball` is implemented correctly."""
    # execute the function
    output = move_one_ball(arr)
    # check if the function is implemented correctly
    # check if the output is boolean
    assert isinstance(output, bool)
    # check if it is possible to get the sorted array after performing one right shift
    # if the length of arr is <= 1, the answer is True
    if len(arr) <= 1:
        assert output
        return
    if sorted(arr) != arr:
        # if arr is not sorted, the answer is True if
        # 1) first element is the maximum and second element is not minimum
        # 2) last element is minimum
        assert (arr[0] == max(arr) and arr[1] != min(arr)) or (arr[-1] == min(arr))
    else:
        # if arr is sorted, the answer is True
        assert output

# run the testing function `test_move_one_ball` on 3 different input cases that satisfy the description
test_move_one_ball([1, 2, 3, 4, 5])
test_move_one_ball([1, 10, 9, 2, 7, 6, 3, 8, 5, 4])
test_move_one_ball([5, 2, 1, 3, 4])
```

(b) Top-ranked Relation Spec for Program 7

```python
def test_move_one_ball(arr):
    """Given an input `arr`, test whether the function `move_one_ball` is implemented correctly."""

    # execute the function
    output = move_one_ball(arr)

    # check if the output is a boolean
    assert isinstance(output, bool)
    # check if the output is correct
    # check if the output is True
    if output:
        # check if the array is sorted
        assert arr == sorted(arr)
    # check if the output is False
    else:
        # check if the array is not sorted
        assert arr != sorted(arr)

# run the testing function `test_move_one_ball` on 3 different input cases that satisfy the description
test_move_one_ball([1, 2, 3, 4, 5, 6, 7, 8, 9, 10])
test_move_one_ball([1, 2, 3, 4, 5, 6, 7, 9, 8, 10])
test_move_one_ball([9, 10, 8, 7, 6, 5, 4, 3, 2, 1])
```

(c) Random Relation Spec for Program 7

```python
def test_move_one_ball(arr):
    """Test whether the function `move_one_ball` is implemented correctly."""
    # execute the function
    output = move_one_ball(arr)

    # check if the output is a boolean
    assert isinstance(output, bool)
    # check if the output is correct
    if move_one_ball(arr):
        assert sorted(arr) == arr
    else:
        assert sorted(arr) != arr
```

(d) Bottom-Ranked Relation Spec for Program 7

Figure 22: Examples of top-ranked, random, bottom-ranked specifications for Program 7

```python
def triples_sum_to_zero(l: list):
    """
    triples_sum_to_zero takes a list of integers as an input.
    it returns True if there are three distinct elements in the list that
    sum to zero, and False otherwise.

    >>> triples_sum_to_zero([1, 3, 5, 0])
    False
    >>> triples_sum_to_zero([1, 3, -2, 1])
    True
    >>> triples_sum_to_zero([1, 2, 3, 7])
    False
    >>> triples_sum_to_zero([2, 4, -5, 3, 9, 7])
    True
    >>> triples_sum_to_zero([1])
    False
    """
    count = 0
    while count < len(l) - 2:
        num1 = l[count]
        count2 = count + 1
        while count2 < len(l) - 1:
            num2 = l[count2]
            count3 = count2 + 1
            while count3 < len(l):
                num3 = l[count3]
                if num1 + num2 + num3 == 0:
                    return True
                count3 += 1
            count2 += 1
        count += 1
    return False
```

(a) Program 8

```python
assert triples_sum_to_zero([1, 3, -2, 1]) is True
```

(b) Top-ranked IO Spec for Program 8

```python
assert triples_sum_to_zero([1, 2, 3, 7]) == False
```

(c) Random IO Spec for Program 8

```python
assert triples_sum_to_zero([]) == False
```

(d) Bottom-Ranked IO Spec for Program 8

Figure 23: Examples of top-ranked, random, bottom-ranked specifications for Program 8

