# OpenReview forum: "Toward Trustworthy Neural Program Synthesis"
_ICLR.cc/2025/Workshop/BuildingTrust — Submitted to BuildingTrust_

### Official Review · Reviewer_EY8D · 2025-02-23

**Rating:** 5
**Confidence:** 4

**Review:**

The paper presents a novel approach to improve the trustworthiness of neural program synthesis. It introduces a system that estimates the probability of a generated program being correct, using both candidate programs and candidate specifications (predicates). This method allows for well-calibrated probabilistic predictions of program correctness. In addition, the paper emphasizes explainability by generating human-readable specifications that clarify program behavior, making the system more interpretable. The results show that the proposed method not only enhances the trustworthiness and transparency of the generated programs but also maintains state-of-the-art generation accuracy. The approach addresses issues of calibration, explainability, and accuracy, providing a robust framework for program synthesis using large language models.

Strengths
- The method provides calibrated probabilistic predictions of program correctness, improving the reliability of generated programs.
- By generating human-readable specifications, the approach enhances the interpretability of neural program synthesis, making the system more transparent.
- The framework maintains high generation accuracy while addressing calibration and explainability, setting a new benchmark in program synthesis.

Weaknesses
- The system's reliance on generating human-readable specifications and probabilistic predictions adds complexity to the program synthesis process.
- While effective for certain tasks, the approach may face scalability issues when applied to more complex or larger programs.
- The method's performance relies on the availability and accuracy of candidate specifications (predicates), which may limit its applicability in some scenarios.

---

### Official Review · Reviewer_TgoD · 2025-02-28
**Good direction but poor evals and lots of questions about generalizability**

**Rating:** 3
**Confidence:** 5

**Review:**

Love the idea but have questions about generalizability
This work takes a look at code generation using LLMs but through the lens of formal methods, specifically specification formulation. The authors leverage older codex models to implement an approach where the LLMs are asked to complete programs, input-output specifications, and relation specifications (akin to fuzzing test cases). They train a single layer classifier to determine if the generated outputs are probably correct.

Pros:
- A novel idea which blends generative AI with program synthesis
- A two pronged approach to program synthesis is
- Good selection of metrics and their definitions were clearly articulated
- Good selection of a relevant dataset
- Good mix of metrics and human evals

Cons:
- The work was rushed. The following are clues as to why:
  - Grammatical errors through out the paper
  - Some references are improperly used. $\texttt{swebench}$ is listed as an LLM system to solve GitHub issues. It's not; it's a benchmark
  - Security concerns are dropped in the conclusion but nothing was previously mentioned
- The classifier is trained over data samples from two datasets but generalizability past this was not evaluated. This implies, their method is not scalable and for each distribution, a classier trained over lots of samples would be needed.
- I don't understand why the following statement holds, "logical relations could serve the long tail of novel tasks that the LLM cannot reliably predict outputs for"
- An ablation into the learned weights of the classifier is missing. It would have been extremely insightful to see what components actually have the most indication for whether a solution is correct.
- Without clear ablations and generalizability results, it's hard to justify this paper which standards in contrast to peer reviewed results which directly combat this (see Syzygy published at LLM4Code 2025)

---

### Official Review · Reviewer_At7S · 2025-03-02
**Paper incomplete and not reflective of SOA**

**Rating:** 3
**Confidence:** 5

**Review:**

Although the premise of the paper has promise, it appears this paper is in an unfinished state. The abstract has a typo, and the introduction appears to be unfinished.
Additionally, the results reported are not meaningful given the current state of the art: HumanEval and MBPP are used instead of SWEBench, and the models used are davinci and cushman, which have both been deprecated for quite some time in favor for much better models.

---

### Decision · Program_Chairs · 2025-03-04

**Decision:**

Reject

**Comment:**

This works is a bit rushed with several errors and lack of substantial experiments. The reviewers have provided several suggestions to strengthen the paper.